# TAFRO Syndrome: A Syndrome or a Subtype of Multicentric Castleman Disease?

**DOI:** 10.3390/biomedicines12030652

**Published:** 2024-03-14

**Authors:** Kazue Takai

**Affiliations:** Department of Hematology, Niigata City General Hospital, Niigata 950-1197, Japan; k-takai@koutouen.jp

**Keywords:** TAFRO syndrome, idiopathic multicentric Castleman disease (iMCD), iMCD-TAFRO, idiopathic plasmacytic lymphadenopathy with polyclonal hyperimmunoglobulinemia (IPL)

## Abstract

TAFRO (thrombocytopenia, anasarca, fever, reticulin fibrosis of bone marrow/renal dysfunction, organomegaly) syndrome is a systemic inflammatory disorder of unknown etiology. It has been recognized as a subtype of idiopathic multicentric Castleman disease (iMCD), and the international diagnostic criteria for iMCD-TAFRO require a lymph node histopathology consistent with iMCD. Furthermore, TAFRO syndrome is defined as a heterogeneous clinical entity caused by underlying diseases such as malignancy, autoimmune diseases, or infections. However, the cases that led to the proposal of TAFRO syndrome lacked recognizable lymphadenopathy and were inconsistent with any other diseases, despite vigorous efforts in differential diagnosis. Irrespective of the presence or absence of Castleman disease (CD)-like histology, TAFRO syndrome exhibits homogeneous clinical, laboratory, and prognostic features, setting it apart from iMCD without TAFRO syndrome. Defining iMCD-TAFRO apart from TAFRO syndrome is deemed meaningless and confusing. MCD is a heterogeneous lymphoproliferative disorder consisting of several subtypes with different pathogenesis, clinical manifestations, and histological features. Typical MCD in Japan, characterized by the histology of plasma cell type and marked polyclonal hypergammaglobulinemia, is identical to idiopathic plasmacytic lymphadenopathy with polyclonal hyperimmunoglobulinemia (IPL). Although IPL is classified into iMCD-NOS (not otherwise specified), it should be recognized as a distinct clinicopathological entity. Furthermore, we propose to separate TAFRO syndrome from the MCD category as a defined disorder.

## 1. Introduction

TAFRO syndrome is a systemic inflammatory disorder characterized by a constellation of symptoms: thrombocytopenia with reticulin fibrosis of the bone marrow, anasarca, fever, renal dysfunction, and organomegaly (hepatosplenomegaly and lymphadenopathy) [1]. Because the lymph node histology of TAFRO syndrome often resembles the mixed type of multicentric Castleman disease (MCD), it has been considered as a subtype of idiopathic MCD (iMCD) [2,3]. It is also described as a severe form of iMCD in international treatment guidelines [4]. However, the first two cases that led to the proposal of TAFRO syndrome did not exhibit lymphadenopathy and could not be diagnosed as MCD [5]. Furthermore, the introduction of the term iMCD-TAFRO, apart from TAFRO syndrome as a subtype of iMCD, has contributed to confusion regarding the concept of TAFRO syndrome [6,7].

Throughout its historical course, Castleman disease (CD) has encompassed various diseases with a broad spectrum of clinical presentations, diverse etiologies, and histopathologies. Considering the detailed proposal of TAFRO syndrome and the historical background of CD, we aim to explore the relationship between TAFRO syndrome and MCD.

## 2. Historical Background of Castleman Disease

In 1954, Castleman reported the histopathological features of a case exhibiting massive mediastinal lymph node hyperplasia characterized by hyalinized capillaries penetrating the follicles [8]. This condition was later termed Castleman disease (CD), hyaline vascular (HV) type. In 1969, Flendrig described the plasma cell (PC) type of CD, where plasma cells proliferate in sheets between follicles [9]. 

In 1980, Mori et al. reported 10 cases of idiopathic plasmacytic lymphadenopathy with polyclonal hyperimmunoglobulinemia (IPL) [10]. This condition was characterized by marked polyclonal hyperimmunoglobulinemia and generalized superficial lymphadenopathy with severe plasma cell proliferation. The histology closely resembled that of PC type CD, except for the systemic involvement of lymph nodes.

In 1983 and 1985, Frizzera et al. reported 15 cases of systemic lymphoproliferative disorder with morphologic features of CD, which they named multicentric CD (MCD) [11,12]. They illustrated the diverse histology of MCD, outlining PC type, HV type, and intermediate histology (mixed type). Consequently, CD was categorized into unicentric CD (UCD) and MCD, with the latter further classified into three histologic types. 

Since the 1980s, numerous cases of Kaposi’s sarcoma and MCD in human immunodeficiency virus (HIV)-infected patients have been reported in Europe and the U.S. [13]. In 1995, it was revealed that this disease was caused by human herpesvirus (HHV)-8 infection [14]. As HHV-8-positive plasmablasts appear in the lymph nodes of HHV-8-associated MCD, it is histologically classified as plasmablastic type [15].

In 2008, Kojima et al. analyzed 28 Japanese cases with HHV-8-negative MCD clinicopathologically, classifying them into two types: IPL (*n* = 18) and non-IPL type (*n* = 10) [16]. IPL resembled PC type MCD in Western countries, characterized by prominent polyclonal hypergammaglobulinemia. The non-IPL type was characterized by mixed type or HV type histology, a high incidence of pleural effusion and ascites, and was often associated with autoimmune disease during the course. Consequently, they mentioned that a portion of non-IPL type might be secondary MCD (autoimmune-disease-associated LPD).

In Western countries, HHV-8-positive MCD in HIV patients was common, and HHV-8-negative MCD had not received much attention. However, in 2014, Fajgenbaum et al. found that more than half of the clinical cases of MCD were HHV-8-negative, naming it idiopathic MCD (iMCD) [17]. 

## 3. Story of TAFRO Syndrome Proposal

### 3.1. Case Presentation

The following are brief descriptions of the first three cases to aid in understanding the characteristics of TAFRO syndrome. 

#### 3.1.1. Case 1

A 47-year-old woman was admitted due to a persistent high fever that did not respond to antibiotic treatment. Progressive symptoms included severe thrombocytopenia, edema, massive pleural effusion, and ascites. A computed tomography (CT) scan revealed hepatosplenomegaly but no lymphadenopathy. Laboratory findings indicated elevated levels of alkaline phosphatase (ALP) and C-reactive protein (CRP), along with severe hypoalbuminemia. The antinuclear antibody (ANA) was positive, but other disease-specific autoantibodies, including the anti-DNA antibody, were all negative. A bone marrow biopsy revealed megakaryocyte proliferation and reticulin fibrosis without malignant cells. Treatment with high-dose corticosteroid (CS) led to a slow improvement in symptoms. Despite persistent elevated ALP levels, cholecystectomy and liver biopsy procedures showed a non-specific liver histology without lymphoma or primary biliary cirrhosis. The patient remained CS-dependent, experiencing repeated relapses with tapering of CS.

#### 3.1.2. Case 2

A 56-year-old man was admitted with thrombocytopenia and progressive edema. The diagnosis of rapidly progressive glomerulonephritis was made due to fever, renal dysfunction with proteinuria, and a high level of CRP. CS pulse therapy and high-dose immunoglobulin therapy were ineffective for severe thrombocytopenia and renal dysfunction with oliguria. The patient required frequent platelet transfusions and hemodialysis. A CT scan showed pleural effusion, ascites, and mild hepatosplenomegaly but no significant lymphadenopathy. Splenectomy and biopsies of the liver and peritoneum revealed non-specific histological findings. A bone marrow biopsy showed increased megakaryocytes and reticulin fibrosis. Immunosuppressive therapy with cyclosporin A slowly improved his symptoms, ultimately leading to complete remission.

#### 3.1.3. Case 3

A 49-year-old man was referred to our hospital due to a high fever, pleural effusion, ascites, and severe thrombocytopenia. Laboratory findings showed severe hypoalbuminemia and elevated levels of ALP and CRP, but all specific autoantibodies examined were negative. A bone marrow biopsy revealed megakaryocyte proliferation and fine reticulin fibrosis. A CT scan showed massive pleural effusion, ascites, mild hepatosplenomegaly, and systemic lymphadenopathy less than 1 cm in diameter. Histological examination of the right inguinal lymph node revealed paracortical hyperplasia with vascular proliferation and atrophic germinal centers resembling the histology of HV type CD. But, his clinical manifestations were completely different from MCD, which usually shows chronic inflammatory symptoms. Despite CS pulse therapy and high-dose immunoglobulin therapy, the patient succumbed to multiple organ failure 42 days after admission.

The autopsy revealed disseminated cytomegalovirus (CMV) infection and marked hemophagocytic histiocytosis, assumed to be a complication of immunosuppressive therapy. There were no findings of malignant lymphoma or other specific findings.

#### 3.1.4. Summary of Three Cases

All three cases showed fever refractory to antibiotics, severe thrombocytopenia, progressive edema, pleural effusion, and ascites. CT scans revealed mild hepatosplenomegaly with or without small lymphadenopathy. Increased megakaryocytes and mild reticulin fibrosis of the bone marrow was shown in the three patients. Histological examinations of the liver and spleen revealed non-specific findings and did not indicate malignant lymphoma. Only one case exhibited CD-like histology upon lymph node biopsy due to inconspicuous lymphadenopathy. Common laboratory findings in the three cases included elevated levels of CRP and ALP and severe hypoalbuminemia with normal immunoglobulin level. The disease-specific autoantibodies examined were all negative.

Despite extensive efforts in differential diagnosis, the constellation of symptoms and laboratory findings did not match any known autoimmune diseases or well-defined lymphoproliferative disorders (LPD). In 2010, we reported these cases as TAFRO (thrombocytopenia, anasarca, fever, reticulin fibrosis of bone marrow, organomegaly) syndrome (tentative term), in order to initiate discussions on whether this condition might represent a novel systemic inflammatory disorder [5].

### 3.2. Proposal of TAFRO Syndrome 

Following the initial report of TAFRO syndrome, several similar cases emerged within Japan. We also identified two new cases and reported the clinical and laboratory features of five cases (including the initial three cases) [18]. Since neither of the new patients exhibited evaluable lymphadenopathy, lymph node histology was obtained from only one of the five cases. Two of the five patients needed hemodialysis. In order to define the concept of TAFRO syndrome more precisely, a research meeting was organized in 2012. Renal dysfunction in TAFRO syndrome often manifests as progressive renal insufficiency, necessitating hemodialysis. Consequently, “R” of the acronym TAFRO was adopted to represent renal dysfunction instead of reticulin fibrosis [19]. TAFRO syndrome was defined as a systemic inflammatory disorder of unknown etiology occurring in patients without any known autoimmune diseases or other well-defined LPD. 

The research meeting evolved into the Japan TAFRO syndrome Research Group under the Research Program for Intractable Disease by Ministry of Health, Labor and Welfare (MHLW) Japan in 2015. The group proposed diagnostic criteria, a disease severity classification, and a treatment strategy for TAFRO syndrome [20], which were subsequently updated in 2019 [21]. Diagnosis of TAFRO syndrome necessitates the presence of all three major categories: anasarca, thrombocytopenia, and systemic inflammation (defined as fever and/or elevated CRP), along with at least two of four minor categories. Mild lymphadenopathy with CD-like histology is a common feature of this syndrome. However, in some cases, lymph node biopsies may be challenging due to anasarca, bleeding tendencies, or the absence of evaluable lymphadenopathy. Therefore, these diagnostic criteria define lymph node histopathology as a minor category, with a strong recommendation for lymph node biopsy, if feasible, to exclude malignancies. Since clinical symptoms and laboratory findings of TAFRO syndrome are non-specific and lack specific serological markers, it is essential to exclude other diseases that can cause similar symptoms. Based on the diagnostic criteria, the Japanese TAFRO syndrome Research Team conducted a nationwide case registry of TAFRO syndrome and MCD, along with a clinicopathological analysis. 

## 4. Relationship between TAFRO Syndrome and MCD

In 2016, Iwaki et al. identified 25 cases of TAFRO syndrome through a review of clinicopathological findings in cases previously diagnosed with MCD [2]. These patients with TAFRO syndrome had an aggressive clinical course. Due to the distinct difference in histology and clinical features between cases with and without TAFRO syndrome, they divided iMCD into two subtypes: TAFRO-iMCD and iMCD-NOS (not otherwise specified). iMCD-NOS showed polyclonal hypergammaglobulinemia and histologic features characteristic of PC type MCD. 

The diagnostic criteria for TAFRO-iMCD, as proposed by Iwaki et al. in 2016, necessitate lymph node histopathology consistent with iMCD [2]. Therefore, TAFRO syndrome lacking lymphadenopathy cannot be diagnosed according to their criteria. Their TAFRO-iMCD cases were selected from previously diagnosed iMCD, with a primary focus on the iMCD category. 

In 2017, Fajgenbaum et al. introduced international consensus diagnostic criteria for iMCD, requiring both major criteria: histopathologic lymph node features consistent with the iMCD spectrum and enlarged lymph nodes in multiple stations. Histologically, iMCD was classified into three types: hypervascular, mixed, and plasmacytic pathology [22]. They observed that iMCD patients with TAFRO syndrome frequently exhibit hypervascular or mixed pathology. The minor criteria comprise six laboratory criteria and five clinical criteria, requiring at least two (including one laboratory criterion). The laboratory criteria are common and non-specific, encompassing both thrombocytopenia and thrombocytosis. Therefore, exclusion criteria are crucial for iMCD, as in the diagnosis of TAFRO syndrome.

In the 2021 international definition of iMCD–TAFRO proposed by Nishimura et al., cases with histology consistent with iMCD are labeled definite iMCD-TAFRO. Those lacking lymph node histology are deemed probable iMCD-TAFRO, and cases with lymph node histology inconsistent with iMCD or meeting exclusion criteria are classified as TAFRO syndrome [6,7]. However, it is imperative in the diagnosis of TAFRO syndrome to rule out diseases listed in the exclusion criteria [20,21]. Nonetheless, in their definition, TAFRO syndrome is described as a heterogeneous clinical entity caused by underlying diseases such as malignancy, autoimmune diseases, or infection [6,7]. This misconception might hinder accurate diagnosis and appropriate therapeutic approaches for both TAFRO syndrome and underlying diseases. 

Lymph node biopsy, if feasible, is strongly recommended in TAFRO syndrome to exclude malignant lymphoma [20,21]. Characterizing lymph node histology in TAFRO syndrome is important for understanding the pathophysiology, and the distinctive histology aids in confirming the diagnosis [23].

Nevertheless, an analysis of registered cases in Japan revealed that approximately 20% of TAFRO syndrome cases have been diagnosed without lymph node biopsies, and even when biopsied, these lymph nodes were small measuring less than 1.5 cm in maximum diameter. Furthermore, whether with or without CD-like histopathology, TAFRO syndrome shares common clinical, laboratory, and prognostic features [24]. Hypervascular or mixed-type histopathologic features are not exclusive to iMCD-TAFRO and can be found in various inflammatory, autoimmune, infectious, and neoplastic diseases [17]. Distinguishing iMCD-TAFRO from TAFRO syndrome is deemed unimportant, as there is no difference in the therapeutic approach. 

POEMS (polyneuropathy, organomegaly, endocrinopathy, M-protein, skin change) syndrome showing CD-like histopathology is classified as POEMS-associated MCD [22]. However, a lymph node biopsy is not mandatory for the diagnosis of POEMS syndrome [25], and the presence or absence of biopsy-proven CD does not significantly impact the treatment strategy and prognosis [26,27]. 

Typical MCD in Japan, characterized by the histology of PC type, marked polyclonal hypergammaglobulinemia, and chronic inflammatory symptoms, is consistent with IPL proposed by Mori et al. [10,16,28]. Dysregulated overproduction of interleukin-6 (IL–6) is implicated in the pathogenesis of PC type MCD [29], and anti-IL-6 receptor antibody is highly effective for MCD in Japan [30]. IPL is considered a distinct clinicopathological entity [28], and the term IPL is more appropriate than iMCD-NOS or PC type iMCD. Personally, I believe that IPL should be separated from the iMCD category as a distinct disease entity. If IPL is still classified into the iMCD category [31], it should be called iMCD–IPL, not iMCD-NOS.

Throughout its history, MCD has encompassed a wide variety of diseases based on similarities in lymph node histology [28,32]. We propose that disorders with a defined concept, such as POEMS syndrome [25] and TAFRO syndrome [20,21], should be separated from the MCD category. Cases with histologically diagnosed MCD that do not consist of IPL or any other diseases should be treated as non-IPL or iMCD-NOS (Table 1). Some cases of iMCD showed positive autoimmune antibodies, and some cases of non-IPL were diagnosed as having an autoimmune disease during the follow-up [16]. This suggests that some non-IPL type MCD cases may be undiagnosed or ill-defined autoimmune diseases [33], as TAFRO syndrome had masqueraded as iMCD before the proposal of TAFRO syndrome.

## 5. Conclusions

TAFRO syndrome has been viewed as a subtype of iMCD, with the international diagnostic criteria for iMCD-TAFRO requiring lymph node histopathology [7]. There is a prevailing misconception that TAFRO syndrome represents a heterogeneous clinical entity triggered by underlying conditions like autoimmune diseases or infections [7]. However, irrespective of the presence or absence of CD-like histopathology, TAFRO syndrome exhibits homogeneous clinical, laboratory, and prognostic features, setting it apart from iMCD without TAFRO syndrome [24]. The accurate differential diagnosis of diseases with similar symptoms to TAFRO syndrome is essential to determine appropriate treatments for each condition.

MCD is a diverse lymphoproliferative disorder comprising various subtypes characterized by distinct pathogenesis, clinical manifestations, and histological features [28,32]. Although IPL is classified as iMCD-NOS, it should be recognized as a distinct clinicopathological entity [28].

In the future, it is expected that identification of a highly specific biomarker for TAFRO syndrome will enable diagnosis without the need for lymph node biopsies. Furthermore, advancements in molecular biological research are anticipated to elucidate the pathogenesis and pathophysiology of TAFRO syndrome, thereby contributing to the evolution of effective treatments.

## Figures and Tables

**Table 1 biomedicines-12-00652-t001:** Diseases that manifest with Castleman-disease-like histology.

1. Unicentric Castleman disease
2. Multicentric Castleman disease (MCD)
HHV-8-associated MCD
Idiopathic MCD
IPL (iMCD-IPL)
Non-IPL (iMCD-NOS)
3. Diseases to differentiate
POEMS syndrome
TAFRO syndrome
Autoimmune diseases (i.e. SLE)
Adult-onset Still disease
IgG4-related diseases
Infectious diseases (i.e., EBV, HIV and TB)
Malignant lymphoma

IPL: idiopathic plasmacytic lymphadenopathy with polyclonal hyperimmunoglobulinemia. NOS: not otherwise specified. SLE: systemic lupus erythematosus. EBV: Epstein-Barr virus. HIV: human immunodeficiency virus. TB: tuberculosis. (Modified from Ref. [1] Takai. K.).

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
