# Peer review of "TAFRO Syndrome: A Syndrome or a Subtype of Multicentric Castleman Disease?"

_biomedicines, 2024, doi:10.3390/biomedicines12030652_

Round 1

Reviewer 1 Report

Comments and Suggestions for Authors

The presented manuscript entitled "TAFRO syndrome and Multicentric Castleman Disease: Proposal of TAFRO Syndrome and Historical Background of Castleman disease" as viewpoint seems to be very interesting and it is very well written though I would suggest to pay attention on the use of the acronyms, because generally it is not recommended to use the acronym in the abstract and even worst in the keywords. It should be highlighted that the acronym should be presented just one time with the detailed words and later, it is recommended to use only the acronym. The content is very informative and all the presented citations and references are updated and sufficient for this type of paper. I would suggest that this manuscript can be accepted after minor revision.

Author Response

Thank you very much for taking time to review this manuscript and giving valuable advice on the use of the acronyms. Please find the detailed responses below and the corresponding revisions highlighted in red in the re-submitted files.

Abstract; TAFRO ( p1. L6-7), CD (p1. L15, 17), Introduction; iMCD (P1. L38)

Body; TAFRO(p3. L106), HV, PC(P4), HIV(p4, L156), HHV(P4. L158), POEMS(p5.L219)

Reviewer 2 Report

Comments and Suggestions for Authors

The manuscript (viewpoint) entitled "TAFRO syndrome and Multicentric Castleman Disease:  Proposal of TAFRO Syndrome and Historical Background of Castleman disease" presents an important overview of the TAFRO syndrome regarding the difficulty of diagnosis and its possible relation with the MCD. The authors nicely presented some cases that have different clinical and pathological features like inconstant lymph node histology. They also emphasized the role of lymph node histopathology reports in the diagnosis of TAFRO. They also highlighted the importance of future molecular diagnostic markers to distinguish TAFRO syndrome from other MCDs. The manuscript is interesting for inflammatory disease readers and researchers.

Author Response

Thank you very much for taking the time to review this manuscript, and for your understanding in it. Please find the detailed responses below and the revisions highlighted in red in the re-submitted files.

We corrected some of the description of the case presentations and the summaries of these cases at the end of the case presentation.

p2. L48-49,  L69-70, L74-75, L87-88,  P3. L94-108

Reviewer 3 Report

Comments and Suggestions for Authors

The article is interesting but very confusing. The document provides a comprehensive overview of TAFRO syndrome and its categorization offering valuable information for medical professionals. However, it is not well supported and is written in a way that is not very comprehensible.

The limits of disease categorization need to be well defined.

On the other hand, it supports the discussion in 3 cases. 

I understand that it's a point of view, but if it's based on three cases, it complicates the approach.

The title should be in line with the article.

The author should not be discouraged and, by reformulating the work, he can make it more interesting and appealing.

Author Response

Thank you very much for taking the time to review this manuscript, and for your valuable comments and encouragement to resubmit this paper.

This article is not a comprehensive Review, but rather a Viewpoint on the disease concept of TAFRO syndrome. We are concerned that the current international definition of TAFRO syndrome were developed from the perspective of MCD and deviate from the original clinical perspective. To aid in understanding TAFRO syndrome, we outlined the first three cases, but TAFRO syndrome was proposed by the Japan TAFRO syndrome Research Group. 

We hope you will understand the title of this article, which focused on the proposal of the TAFRO syndrome and the historical background of Castleman disease. We corrected some of the descriptions of case presentations and summarized the cases at the end of the case presentation. Please find the detailed revisions highlighted in red in the re-submitted files.

Reviewer 4 Report

Comments and Suggestions for Authors

Thank you for the submission of this manuscript on the recently described sub-type of iMCD, called TAFRO. The case series and review is very well presented. My only comment is that it would be helpful to present the three patient case descriptions with a table with columns including the components of TAFRO (thrombocytopenia, anasarca, fever, BM fibrosis and organomegaly) also treatment(s) and response to treatment.  

Author Response

Thank you very much for taking the time to review this manuscript, and for your valuable advice. We received comments to summarize the components of TAFRO and treatment of the three cases using Table. Since case presentations are not main focus of the paper, a summary of these cases is described at the end of the case presentation. Treatment was trial-and-error and varied, so is described in each case presentation. We hope you will understand it.

Please find the detailed revisions highlighted in red in the re-submitted files.

Case presentation; P2. L48-49, L69-70, L74-75, L87-88, P3. L94-108

Round 2

Reviewer 3 Report

Comments and Suggestions for Authors

Thank you for your reply. There is no doubt that the article is interesting and nothing better than presenting cases to support the proposal. I still think the title should be changed. How about making it interrogative? For example TAFRO: a syndrome or a subtype of Multicentric Castleman Disease? or something else that gets readers interested in the work?

The authors' explanation in the answer helps to better understand the work. Thank you. But I think it's the structure that makes the paper a bit confusing. We need to read more than one time to uindersans. One idea is to do the Historical background of MCD after the introduction, then the syndrome proposal and then the presentation of the cases. OR the Historical background of MCD, the presentation of the cases and then the proposal of the syndrome.

If I understand the point correctly, the cases are presented to support the syndrome proposal, right? That's what's missing. Organize the work in such a way that it is better understood. 

Thank you

Author Response

Thank you so much for taking your valuable time to review this manuscript again. I appreciate your sincere and constructive advice. 

I couldn't agree more your advice, and changed the title according to your idea; TFRO syndrome: a syndrome or a subtype of multicentric Castleman disease?

I changed the structure of the paper according to your idea; the historical background of Castleman disease after introduction, then case presentation and then the proposal of TAFRO syndrome.

I understand this paper is lengthy and confusing. So I tried to rewrite it briefly, but the attempt ended in failure except Abstract. Please find the detailed revisions highlighted in red.

I'm truly grateful your encouraging advice.